# Single- and duplex TaqMan-quantitative PCR for determining the copy numbers of integrated selection markers during site-specific mutagenesis in *Toxoplasma gondii* by CRISPR-Cas9

Kai Pascal Alexander Hänggeli[1,2], Andrew Hemphill[1]*, Norbert Müller[1], Bernd Schimanski[3], Philipp Olias[4], Joachim Müller[1], Ghalia Boubaker[1]*

**1** Department of Infectious Diseases and Pathobiology, Institute of Parasitology, University of Bern, Bern, Switzerland, **2** Graduate School for Cellular and Biomedical Sciences, University of Bern, Bern, Switzerland, **3** Department of Chemistry, Biochemistry and Pharmaceutical Sciences, University of Bern, Bern, Switzerland, **4** Institute of Animal Pathology, Vetsuisse Faculty, University of Bern, Bern, Switzerland

* ghalia.boubaker@vetsuisse.unibe.ch (GB); andrew.hemphill@vetsuisse.unibe.ch (AH)

## Abstract

Herein, we developed a single and a duplex TaqMan quantitative PCR (qPCR) for absolute quantification of copy numbers of integrated dihydrofolate reductase-thymidylate synthase (*mdhfr-ts*) drug selectable marker for pyrimethamine resistance in *Toxoplasma gondii* knockouts (KOs). The single TaqMan qPCR amplifies a 174 bp DNA fragment of the inserted *mdhfr-ts* and of the wild-type (WT) *dhfr-ts* (*wtdhfr-ts*) which is present as single copy gene in *Toxoplasma* and encodes a sensitive enzyme to pyrimethamine. Thus, the copy number of the *dhfr-ts* fragment in a given DNA quantity from KO parasites with a single site-specific integration should be twice the number of *dhfr-ts* copies recorded in the same DNA quantity from WT parasites. The duplex TaqMan qPCR allows simultaneous amplification of the 174 bp *dhfr-ts* fragment and the *T. gondii 529-bp* repeat element. Accordingly, for a WT DNA sample, the determined number of tachyzoites given by *dhfr-ts* amplification is equal to the number of tachyzoites determined by amplification of the *Toxoplasma 529-bp*, resulting thus in a ratio of 1. However, for a KO clone having a single site-specific integration of *mdhfr-ts*, the calculated ratio is 2. We then applied both approaches to test *T. gondii* RH mutants in which the *major surface antigen* (SAG1) was disrupted through insertion of *mdhfr-ts* using CRISPR-Cas9. Results from both assays were in correlation showing a high accuracy in detecting KOs with multiple integrated *mdhfr-ts*. Southern blot analyses using BsaBI and DraIII confirmed qPCRs results. Both TaqMan qPCRs are needed for reliable diagnostic of *T. gondii* KOs following CRISPR-Cas9-mediated mutagenesis, particularly with respect to off-target effects resulting from multiple insertions of *mdhfr-ts*. The principle of the duplex TaqMan qPCR is applicable for other selectable markers in *Toxoplasma*. TaqMan qPCR tools may contribute to more frequent use of WT *Toxoplasma* strains during functional genomics.

**Data Availability Statement:** All relevant data are within the paper and its Supporting information files.

**Funding:** - AH: Swiss National Science Foundation (grant No. 310030_184662) www.snf.ch - AH: Uniscientia Foundation. www.interadvice.li The funders had no role in study design, data collection and analysis, decision to publish, or preparation of the manuscript.

**Competing interests:** The authors have declared that no competing interests exist.

# Introduction

*Toxoplasma gondii* is an apicomplexan parasite that causes diseases in farm animals with an enormous global economic impact and a high zoonotic potential [1]. In immunocompetent hosts, infection does not have serious consequences, and proliferative tachyzoites differentiate into tissue cyst-forming bradyzoites, which can persist over many years to lifelong without causing any clinical symptoms. However, *T. gondii* is an opportunistic pathogen, and primary infection in pregnant animals and also women can lead to vertical transmission, and result in fetal malformations and/or abortion. In patients undergoing immunosuppression, either by disease or through medical treatment, reactivation of bradyzoites from tissue cysts and re-differentiation into tachyzoites frequently causes serious pathology. Current drugs for toxoplasmosis treatment typically include antifolates using a combination of pyrimethamine–sulfadiazine or trimethoprim–sulfamethoxazole, and pyrimethamine can also be combined with clindamycin, azithromycin, or atovaquone. These treatments are unspecific, frequently result in adverse effects, and clinical failures have been reported [2, 3]. To date, more than 110 years after the first discovery of *T. gondii* [4], there is still a need for identifying drug targets and vaccine candidates, which could be exploited for the development of better preventive or therapeutic options for the management of toxoplasmosis [5, 6]. In this context, functional genomics plays a major role, and gene knockout (KO) in protozoan parasites is the most commonly applied approach [7]. *Toxoplasma* is highly amenable to genetic manipulation and has thus emerged as one of the major apicomplexan model parasites [8].

Gene KO and gene replacement strategies rely on double crossover homologous recombination (HR) using type I or II *T. gondii* KU80 mutants (Δ*ku80*s) as parental strain [9]. The Δ*ku80* parasites are deficient in the non-homologous end-joining (NHEJ) pathway required for repairing DNA double-strand breaks (DSBs) [10, 11]. Genetic manipulation of *T. gondii* WT strains is hindered by the presence of a predominant NHEJ as main DSB repair mechanism [12], which results in enhanced random integration of exogenous genes. Despite the fact that Δ*ku80* background increases the efficiency of targeted mutagenesis in *T. gondii* by HR, defective NHEJ might render parasites prone to accumulate chromosomal aberrations [13] causing genomic instability [14], in particular since *T. gondii* tachyzoites are usually maintained *in vitro* through excessive cycles of proliferation and DNA replication.

For positive selection of *T. gondii* mutant or transgenic strains that have successfully integrated an exogenous DNA coding for a modified dihydrofolate reductase-thymidylate synthase (mDHFR-TS), pyrimethamine (Pyr) is the drug of choice [15–19], since mDHFR-TS confers resistance to Pyr [20]. In the genome of WT *T. gondii*, a single-copy gene coding for DHFR-TS (WT-DHFR-TS) is expressed, but the enzyme is sensitive to Pyr [21]. The mDHFR-TS differs from WT- DHFR-TS by three amino acid substitutions, with two being located in exon 1 (Ser **TC**T → Arg **CG**T and Thr A**C**C → Asn A**A**C) and one in exon 3 (Phe **T**TT → Ser **T**CC) [20–22].

Based on the original method of CRISPR-Cas9 that was successfully implemented for genome editing in *T. gondii* in 2014 [23, 24], many alternative protocols have been developed [9] rendering genetic manipulation of WT strains feasible. This advance was possible because in CRISPR-Cas9 formation of a DSB at a specified genomic site is ensured by the 20-nucleotide guide RNA (gRNA) that binds and guides the Cas9 endonuclease to the defined location [25]. Then, the CRISPR-Cas9-mediated DNA break can be repaired through NHEJ or homology-directed repair (HDR) pathways [9]. Options for delivering CRISPR-Cas9 components into the cell as one- or two-vector or cloning-free approaches are now available [26].

Although CRISPR-Cas9 has significantly improved the efficiency of targeted mutagenesis and/or site-specific insertion of selectable markers in *Toxoplasma* WT strains, the Δ*ku80* parasites have remained the first choice for functional genetic studies [9, 27–29]. For Δ*ku80* strains, CRISPR-Cas9 has allowed to considerably reduce the length of homologous flanking DNA to 40 bps [23]. This has rendered the task of template DNA preparation more simple, since these short homology flanking regions of 40 bps can be incorporated into primers designed for the amplification of the selectable marker [26].

A crucial step during CRISPR-Cas9 is the verification of the KO and the validation of gene edits, which must be carried out prior to further functional investigations. Following the selection of mutant clones by drug treatments, PCR and/or Sanger sequencing are used to verify the DNA sequence of the targeted locus [23, 26]. Subsequently, Western blotting and/ or immunostaining are applied to confirm the loss of gene expression [23]. Nonetheless, off-target effects (OTEs) of CRISPR-Cas9 are often not considered. OTEs resulting from non-specific cleavage by a non-complexed Cas9 are of low probability, since endonuclease activity of Cas9 is dependent on the interaction with the gRNA [30] as revealed by crystallographic studies [31–35]. However, a gRNA-independent endonuclease activity by Cas9 in the presence of manganese ions was reported [36]. Overall, the gRNA and the protospacer adjacent motif (PAM) next to the targeted genomic sequence play a crucial role in determining the specificity of gene targeting by CRISPR-Cas9 [37]. For example, the *Streptococcus pyogenes* Cas9 (*Sp*Cas9) binds optimally to a consensus NGG canonical PAM [38–40], but it can also interact, albeit with less affinity, with other non-canonical PAMs [41] mostly NAG and NGA [42, 43]. Furthermore, Cas9 can unspecifically cleave a DNA sequence with up to seven mismatch base pairs in the PAM proximal region of the gRNA sequence known as "seed sequence" [44, 45]. In addition, in the mammalian genome, DNA or RNA bulges, caused by small insertions or deletions, were identified as potential off-target sites [46]. The incidence of off-target mutations by CRISPR-Cas9 widely varies between cell types and species [37], particularly in cells with defective DSB repair pathways [47]. Substantial efforts have been made to develop *in silico* systems for optimal gRNA design. However, prediction and scoring by the algorithms employed are mostly based on DNA-binding rather than cleavage, and even more significant factors such as PAMs, DNA/RNA bulges and experimental conditions are excluded [48].

Whole genome sequencing (WGS) is the only unbiased and direct approach allowing a comprehensive analysis of OTEs including single-nucleotide polymorphisms, indels and other structural differences. However, this approach is costly and time consuming, thus cannot be applied as a first-line testing strategy [37]. Moreover, when the designed strategy to achieve gene KO by CRISPR-Cas9 consists in disrupting the targeted sequence followed by insertion of a selectable marker, it is important to check KO cells for unintended additional integration events. For that, Southern blotting (SB) can be applied, which allows to determine the copy number of inserted exogenous DNA. However, SB requires a relatively large amount of DNA, special equipment, and is relatively time-consuming when many clones have to be analyzed. In addition, the accuracy of SB depends largely on the use of appropriate restriction enzymes.

An alternative strategy to determine single or multiple transgene integration events caused by CRISPR-Cas9 is real-time PCR-based quantification (RT-qPCR), which allows a more high-throughput determination of transgene copy numbers and respective integration patterns (single or multiple insertions) [49–52].

In this study, we aimed at improving the selection protocol for *T. gondii* KO transfectants generated by CRISPR-Cas9, with regard to the identification of OTEs resulting from multiple insertion of selectable marker by developing two TaqMan qPCR-based approaches.

## Materials and methods

### Parasite and cell culture

Tachyzoites of *T. gondii* type I RH strain were maintained *in vitro* in human foreskin fibroblasts (HFF) as previously described [53].

### CRISPR-Cas9 compounds and *mdhfr-ts* selection cassette

The DNA sequence coding for *T. gondii* RH SAG1 was retrieved from GenBank under the accession number GQ253075.1 and used for the design of the 23-nt gRNA (Table 1).

The plasmid P926 encodes a GFP-tagged Cas9 endonuclease and a pre-existing gRNA. The expression of Cas9 is under the control of the bacterial T7 promotor while transcription of the gRNA is driven by the *T. gondii* U6 promotor [23]. The pre-existing gRNA in the plasmid P926 was replaced a by the newly designed 23 nucleotide DNA sequence using site-directed mutagenesis (New England Biolabs, M0491S). Primers are listed in Table 1. The modified P926 plasmid was then amplified in NEB 5-alpha competent *Escherichia coli* (*E. coli*), purified using ZymoPURE Plasmid Miniprep Kit (Zymo Research) and sequenced. The template plasmid P972 was used for amplification of the selectable marker *mdhfr-ts*.

### Transfection and selection

The transfection procedure was adapted from Sidik et al. [23]. Briefly, the electroporation reaction was prepared in a final volume of 300 μL cytomix buffer containing 7.5 μg P926, 1.5 μg of *mdhfr-ts*, $0.112 \times 10^7$ *T. gondii* RH WT tachyzoites, 2 μM adenosine triphosphate (ATP) and 5 μM L-glutathione in 4 mm gap cuvettes (Axonlab, Baden, Switzerland). Cells were than

**Table 1. Sequence of primers and probes used in this study.**

| Label | Sequence 5'-3' |
|---|---|
| gRNA | GGCAGTGAGACGCGCCGTCACGG |
| Q5 mutagenesis_P926 | |
| F-primer | GGCAGTGAGACGCGCCGTCAGTTTTAGAGCTAGAAATAGC |
| R-primer | AACTTGACATCCCCATTTAC |
| Amplification of mDHFR-TS | |
| F-primer | TCCGTAGATCTAAGCTTCGCCA |
| R-primer | AGTGAGCTGATACCGGAAT |
| *sag1* –genotyping PCR | |
| GBtg12 F | TGTCACATGTGTCATTGTCG |
| GBtg13 R | CAGGTGACAACTTGATTGGCA |
| SouthernBlot_dhfr probe | |
| dhfr probe F | ACATCGAGACCAGGTGTG |
| dhfr probe R | ACGATGTTCAATCTGTCCA |
| Q-PCR | |
| dhfr-F | ATCGGCATCAACAACG |
| dhfr-R | GAATCTCTT GCCGACTGA |
| *DHFRQ-P | Cy5- GTGACAAAAACGACGCCCG -BHQ. |
| 529rpe-F | AGGAGAGATATCAGGACTGTAG |
| 529rpe-R | GCGTCGTCTCGTCTAGATCG |
| 529rpeQ-P | FAM-GAGTCGGAGAGGGAGAAGATGTT-BHQ |

(*) TaqMan probes designed in this study.

electroporated with a pulse generator (ECM830, BTX Harvard Apparatus, Holliston, MA) by applying the following protocol: 1700 V, 176 μs of pulse length, two pulses with 100 ms interval. Transfected tachyzoites were transferred immediately into T25 flasks with confluent HFFs, which were placed in a humidified incubator at 37˚C / 5% $CO_2$. After 24 h cultures were subjected to drug selection by the addition of 3 μM Pyr to the culture medium. Clones were isolated by limiting dilution (0.5 tachyzoites/150 μL medium) and allowed to grow in 96 well plates for 10 days.

## PCR and Sanger sequencing

Genomic DNA from thirty-three clones and WT tachyzoites was extracted using the NucleoSpin DNA RapidLyse kit (Macherey-Nagel) according to the manufacturer's instructions. We further examined the SAG1 locus of the thirty-three clones and the WT parasites by PCR. Amplicons of the WT SAG1 locus were ~216 bp, however, for KO clones with one insertion of the complete MDHFR-TS sequence, the expected amplicon length was 3379 bp (~3400 bp). The diagnostic PCR was performed in 50 μL final volume containing 0.2 mM dNTPs, 0.5 μM of each forward (GBtg12) and reverse primers (GBTg13), Q5 high-fidelity DNA polymerase (1 unit) and Q5 high GC enhancer (1x), and 80 ng of template DNA. The GBtg12 F/ GBtg13 R primer sequences are shown in Table 1. Conditions were as follows: initial denaturation at 98˚C for 3 min, 25 cycles of denaturation at 98˚C for 30 sec, annealing at 58˚C for 30 sec, and elongation at 72˚C for 2 min. The final cycle was followed by extension at 72˚C for 2 min. PCR products were purified using Zymo DNA Clean and Concentrator kit (Zymo Research), 20 ng of purified PCR products were submitted to Sanger sequencing.

## Immunofluorescence assay (IFA)

Immunofluorescence microscopy was done as described previously [54, 55]. Briefly, freshly egressed tachyzoites were isolated from infected HFF cultures, fixed in suspension in PBS / 3% paraformaldehyde, and were allowed to attach to poly-L-lysine-coated coverslips for 20 min at room temperature. To permeabilize cells, coverslips were incubated with pre-cooled methanol / acetone (1:1) solution for 20 min at −20˚C. Then samples were rehydrated and incubated overnight at 4˚C in PBS / 3% bovine serum albumin (BSA) solution to block unspecific binding sites. SAG1 expression was assessed by using anti-SAG1 monoclonal antibody (1:1000) and anti-mouse fluorescein-isothiocyanate (FITC) (1:300). For double stainings, SAG1 labelled parasites were further incubated in polyclonal rabbit anti-Inner Membrane Complex 1 (IMC1) antibody (1:500), and a secondary anti-rabbit tetramethyl-rhodamine-isothiocyanate (TRITC) (1:300). Finally, coverslips were mounted onto glass slides using Vectashield mounting medium containing 4, 6-diamidino-2-phenylindole (DAPI).

## SDS-PAGE and Western blotting

Pellets corresponding to equal numbers of WT or Δ*sag1* tachyzoites were prepared and dissolved in Laemmli SDS sample buffer, which contains β-mercaptoethanol. Cell lysates were then separated by SDS-PAGE. Two SDS-PAGEs were made simultaneously; after electrophoresis, one gel was stained with Coomassie and proteins on the other gel were transferred to nitrocellulose filters. The blot was saturated with blocking solution (5% skimmed milk powder and 0.3% Tween 20 in PBS) for 2 hours at room temperature and then incubated with *T. gondii* anti-SAG1 monoclonal antibody (1:500) overnight at 4˚C. After washing, nitrocellulose membrane was incubated with an alkaline-phosphatase conjugated anti-mouse IgG antibody (1:1000). Lastly, reactive bands were visualized by immersion of the blot in 5-bromo-4-chloro-3-indolyl phosphate (BCIP)/nitro blue tetrazolium (NBT) detection solution.

## Single TaqMan-qPCR

To determine the copy numbers of the inserted *mdhfr-ts* selectable marker in the genome of KO clones, we designed a single TaqMan-qPCR taking advantage of the fact that WT *T. gondii* tachyzoites have a single copy of *dhfr-ts* in their genome (*wt dhfr-ts*). Specific *dhfr* forward and reverse primers (Table 1) were designed to yield a 174 bp fragment of the MDHFR-TS or WT DHFR-TS gene. The TaqMan probe DHFRQ-P (Table 1) contained the Cyanine 5 (Cy5) reporter dye at the 5′ end and Black Hole Quencher (BHQ) fluorescent quencher at the 3′ end.

Freshly egressed tachyzoites from infected cultures were filtered through a 3 µM pore-sized polycarbonate membrane, counted and $10^6$ tachyzoites were used for DNA extraction by NucleoSpin DNA RapidLyse Kit according to the instructions provided by the manufacturer. From each tested WT or KO clone, 3 ng DNA were used as template. DNA quantifications were performed by QuantiFluor double-stranded DNA (dsDNA) system (Promega, Madison, WI, USA). PCR amplification was performed in a total reaction mixture of 10 µL containing 1x SensiFast master mix (Bioline, Meridian Bioscience), 0.5 µM of reverse and forward primers, 0.1 µM of DHFRQ-P probe, 0.3 mM dUTP, and one unit of heat-labile Uracil DNA Glycosylase (UDG) [56].

A Bio-Rad CFX 96 QPCR instrument (Biorad) was used with the following thermal profile: (1) initial incubation of 10 min at 42˚C, followed by (2) denaturation step of 5min at 95˚C and (3) 50 cycles of two-step amplification (10 s at 95˚C and 20 s at 62˚C). Samples were tested in triplicates and a negative control with double-distilled water was included for each experiment. For quantification, two standard curves were made: one was based on the use of a 10-fold serial dilution of the plasmid P972 ranging from $1.29 \times 10^9$ to 1.29 copies / 3 µl, and the other one was based on a 10-fold serial dilution of DNA from WT *T. gondii* RH, with tachyzoite numbers ranging from $7.5 \times 10^5$ to 75 per 3 µL [57].

## Duplex TaqMan-qPCR

In this assay, the number of tachyzoites corresponding to 3 ng DNA and the copy number of the DHFR-TS DNA fragment were assessed simultaneously. Quantification of tachyzoites was achieved by preparation of a *T. gondii* standard curve using 10-fold serial dilutions with parasite concentrations ranging from $7.5 \times 10^5$ to 75 and amplification of a 162 bp region of the *T. gondii* 529 bp repeat element [58]. Amplifications were carried-out in total volume of 10 µL containing 1 x SensiFast master mix (Bioline, Meridian Bioscience), 0.5 µM of each primer set (dhfr-F/R and 529rpe-F/R), 0.1 µM of each probe (DHFRQ-P and 529rpeQ-P), 0.3 mM dUTP, and one unit of heat-labile uracil DNA glycosylase (UDG). From each sample, three ng of DNA were used in the reaction mix. All reactions were run in triplicates and amplifications were carried-out under the same thermal profile used for the single TaqMan-qPCR. The cycle threshold values (CT) were plotted as mean of triplicates against the standard curve values to determine the number of tachyzoites. Parasite concentrations were determined after the calculation of the linear regression equation (y = ax + b), where y = CT; a = curve slope (slope); x = parasite number; and b = where the curve intersects y-axis (y intercept).

## Southern blot

Two Southern hybridizations were carried out on seven Δ*sag1* clones and the WT strain that were tested by qPCRs. One µg of each genomic DNA-sample was digested with the restriction enzymes BsaBI or DraIII for 6 h at 60˚C or 37˚C, respectively. Reaction mixtures were then separated by 0.8% agarose gel electrophoresis containing ethidium bromide. Gels were subjected to depurination (15 min in 0.25 M HCl), denaturation (30 min in 1 M NaCl / 0.5 M NaOH) and neutralization (1 hour in 1 M Tris-HCl, pH 7.5/ 3 M NaCl). Separated DNA

fragments were then transferred onto Hybond membrane (Amersham) by capillary transfer and subsequently stably fixed by UV crosslinking for 10 seconds. For blocking non-specific binding sites, membranes were pre-incubated in hybridization buffer (0.5 M $Na_2HPO_4$, 60 mM $H_3PO_4$, 7% SDS, 1% BSA, 0.9 mM EDTA) for 2 hours at 65°C.

The DHFR probe was generated from the plasmid P972 by PCR with DHFR forward and reverse primers listed in Table 1, gel-purified and radioactively labelled with α-P32-dCTP using the Amersham Megaprime DNA Labeling System. The labeled probe was heat-denatured at 95°C for 3 min and added directly to the pre-hybridized membranes. After overnight incubation at 65°C the membranes were washed 15 minutes each in 1 x SSC, 0.1% SDS and 0.5 x SSC, 0.1% SDS and eventually exposed to Phosphoimager screens for 20 hours.

## Results

### Generation of *T. gondii* RH Δ*sag1* clones by CRISPR/Cas9

After transfection and 10 days *in vitro* culture under Pyr treatment, thirty-three clones, together with WT parasites were genotyped by PCR. As shown in Fig 1, the WT locus produced the expected PCR product of ~216 bp. Five clones, namely *T. gondii* RH Δ*sag1* C18, 23, 30, 31 and 33 exhibited a PCR product in the expected size of more than 3 kb, indicating integration of the selection marker. In other clones such as in *T. gondii* RH Δ*sag1* C6 and C7, PCR amplified a product of ≤ 1000 bp. Thus overall, the efficiency of *sag1* disruption through insertion of the *mdhfr-ts* selectable marker in s*ag1* without homology arms was about 15% (5 / 33).

Direct Sanger sequencing of the obtained PCR products revealed that in *T. gondii* RH Δ*sag1* C18, 23, 30 and 33, *sag1* was disrupted by insertion of complete *mdhfr-ts* sequence, while clone C31 had incorporated a truncated *mdhfr-ts* into *sag1*. For clone C6 and 7, the DSB in the SAG1 gene generated by CRISPR-Cas9 was repaired through NHEJ by insertion of

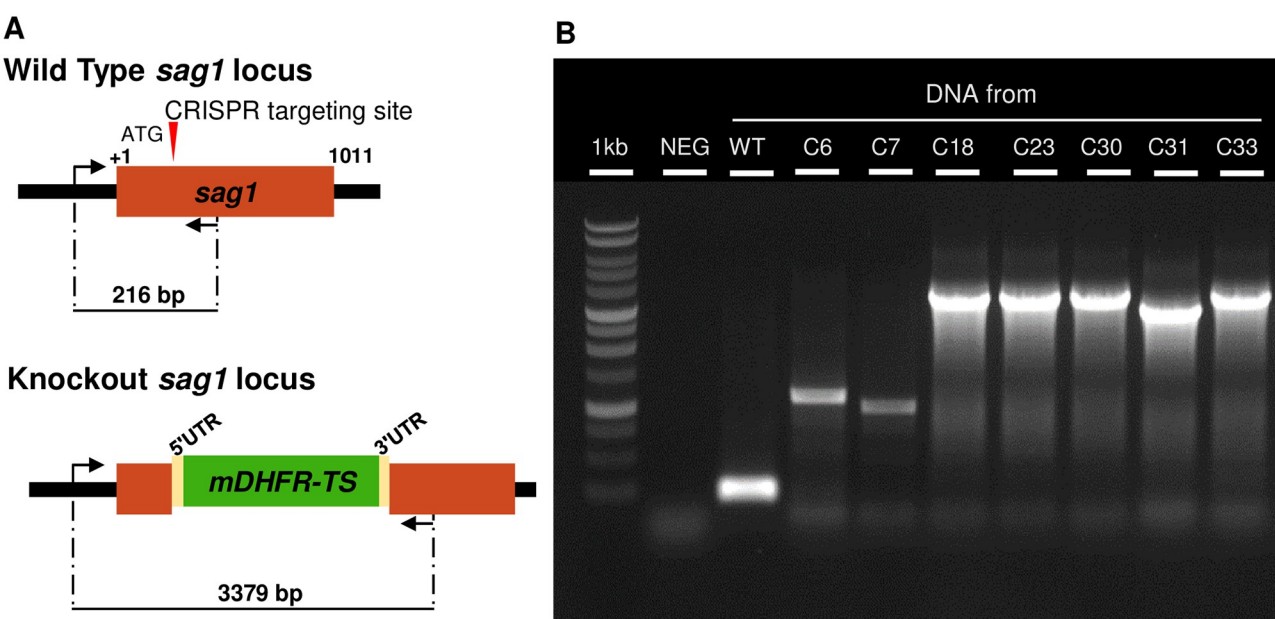

**Fig 1. SAG1 gene disruption in *T. gondii* RH by CRISPR-Cas9 technology. (A)** Schematic representation of the strategy used to disrupt *sag1* by inserting the pyrimethamine-resistance gene MDHFR-TS. (**B**) Diagnostic PCR revealing integration of a complete *mdhfr-ts* sequence into *sag1* in four clones (C18, C23, C30 and C33) compared with the parental strain RH. The KO clone C31 showed a smaller band, clones C6 and 7 exhibited a band ≤ 1000 bp. The WT locus produced the expected PCR product (~ 216 bp).

short DNA sequence (mostly derived from the plasmid P926), while the actual selection marker *mdhfr-ts* was most likely integrated elsewhere in the genome. As shown in Fig 2, Western blot analysis as well as IFA confirmed the absence of TgSAG1 expression in tachyzoites of *T. gondii RH Δsag1* C6, 7, 18, 23, 30, 31 and 33 (Fig 2).

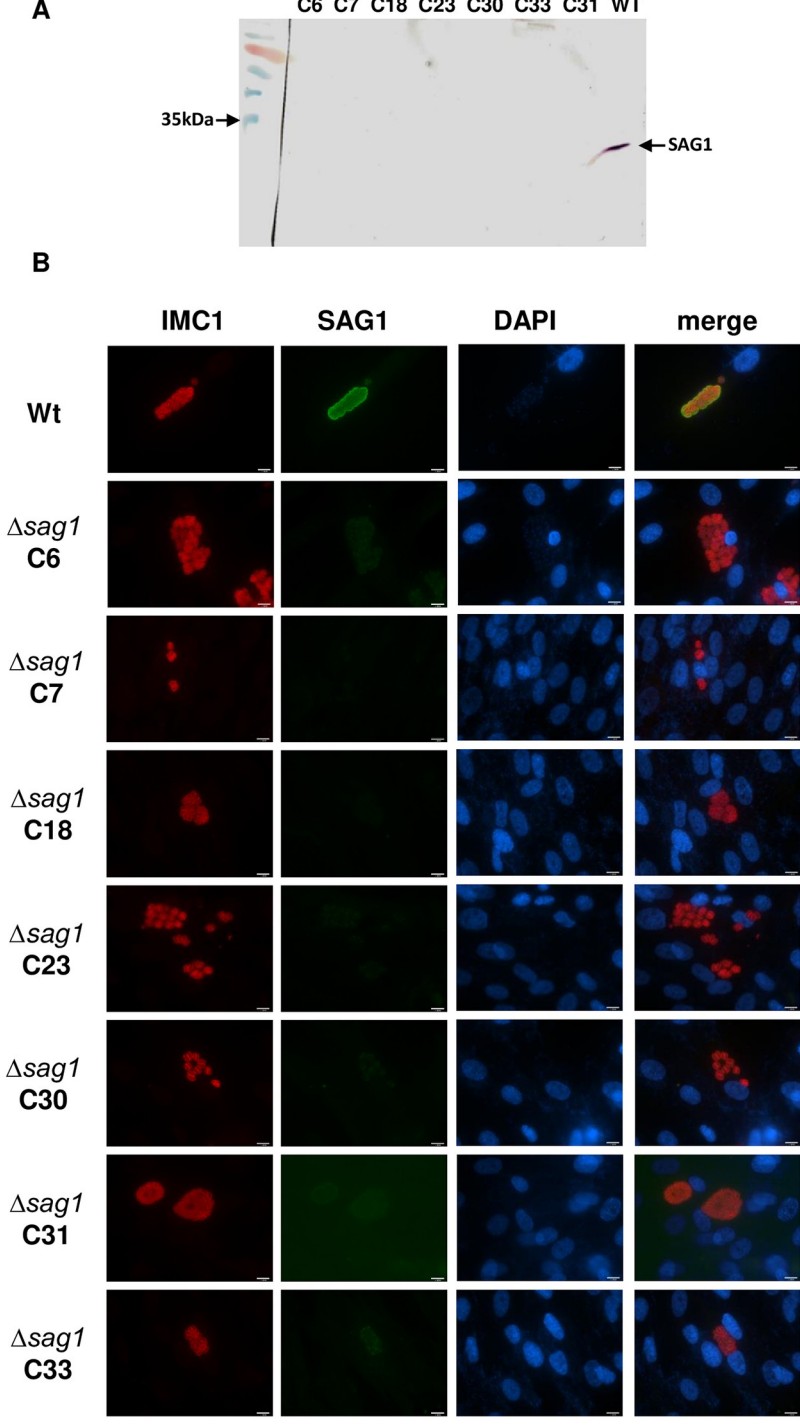

**Fig 2. Loss of *sag1* expression in *T. gondii* RH SAG1 knockouts by (A) Western blot analysis and (B) immunofluorescence.**

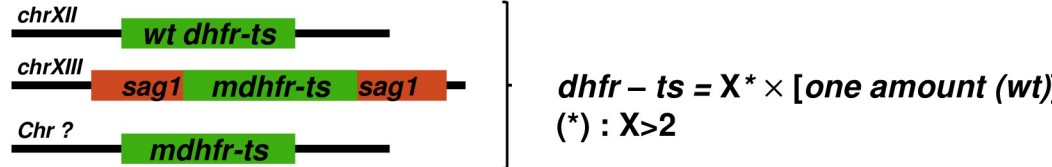

- **_T. gondii_ RH (wt)**

  chrXII    [wt dhfr-ts]    *dhfr – ts = one amount (wt)*

- **_T. gondii_ RH Δ_sag1_ (_single insertion of mdhfr-ts_)**

  chrXII    [wt dhfr-ts]
  chrXIII    [sag1  mdhfr-ts  sag1]    *dhfr – ts = 2 × [one amount (wt)]*

- **_T. gondii_ RH Δ_sag1_ (_multiple insertion of mdhfr-ts >1_)**

  chrXII    [wt dhfr-ts]
  chrXIII    [sag1  mdhfr-ts  sag1]    *dhfr – ts = X* × [one amount (wt)]*
  Chr ?    [mdhfr-ts]    *(\*) : X>2*

**Fig 3. Principle and potential outcomes of the single TaqMan-qPCR.**

## Single TaqMan-qPCR

As shown in Fig 3, this single TaqMan-qPCR aimed to determine whether random integration in *T. gondii RH Δsag1* C18, 23, 30, 31 and 33 occurred elsewhere in the genome beside the detected site-specific integration of *mdhfr-ts* in *sag1*. The principle is based on the fact that the copy number of the *dhfr-ts* fragment in a given DNA quantity of KO parasites with a single site-specific integration should be twice the number of *dhfr-ts* copies recorded in the same DNA quantity from WT parasites (Fig 3).

As shown in Fig 4A and 4B, comparable linear calibrator curves were obtained using serial 10-fold dilutions of *mdhfr-ts* plasmid or *T. gondii* genomic DNA (range $7.5 \times 10^5$ to 75 genome equivalents), indicating thus similar amplification efficiency of *dhfr-ts* from both sources (Fig 4A and 4B).

As shown in Fig 4C, for clones *T. gondii* RH Δ*sag1* C6, 7, 18, 23, 31 and 33, the determined number of *dhfr-ts* copies in the three ng of DNA was almost the double of that number calculated for the WT parasites, independently of the standard curve. The calculated number of inserted *mdhfr-ts* selectable marker was almost equal to 1 for the following clones: *T. gondii* RH Δ*sag1* C6, 7, 18, 23, 31 and 33, as shown in Fig 4D.

## Duplex TaqMan-qPCR

In this assay, quantitative amplification of the *dhfr-ts* and of the *T. gondii* 529-bp repeat element were combined into one reaction (Fig 5).

According to the principle of the duplex TaqMan-qPCR, for WT DNA, the ratio of the number of tachyzoites determined by amplification of *dhfr-ts* to the number of tachyzoites determined by amplification of *T. gondii* 529 bp repeat element is equal to 1. This ratio is equal to 2 or greater than 2 in case of single or multiple insertion of *mdhfr-ts* selection marker, respectively.

The standard curve was made from a 10-fold serial dilution of *T. gondii* RH DNA, with parasite concentrations ranging from $7.5 \times 10^5$ to 75 (Fig 6A). The two primer pairs in the duplex

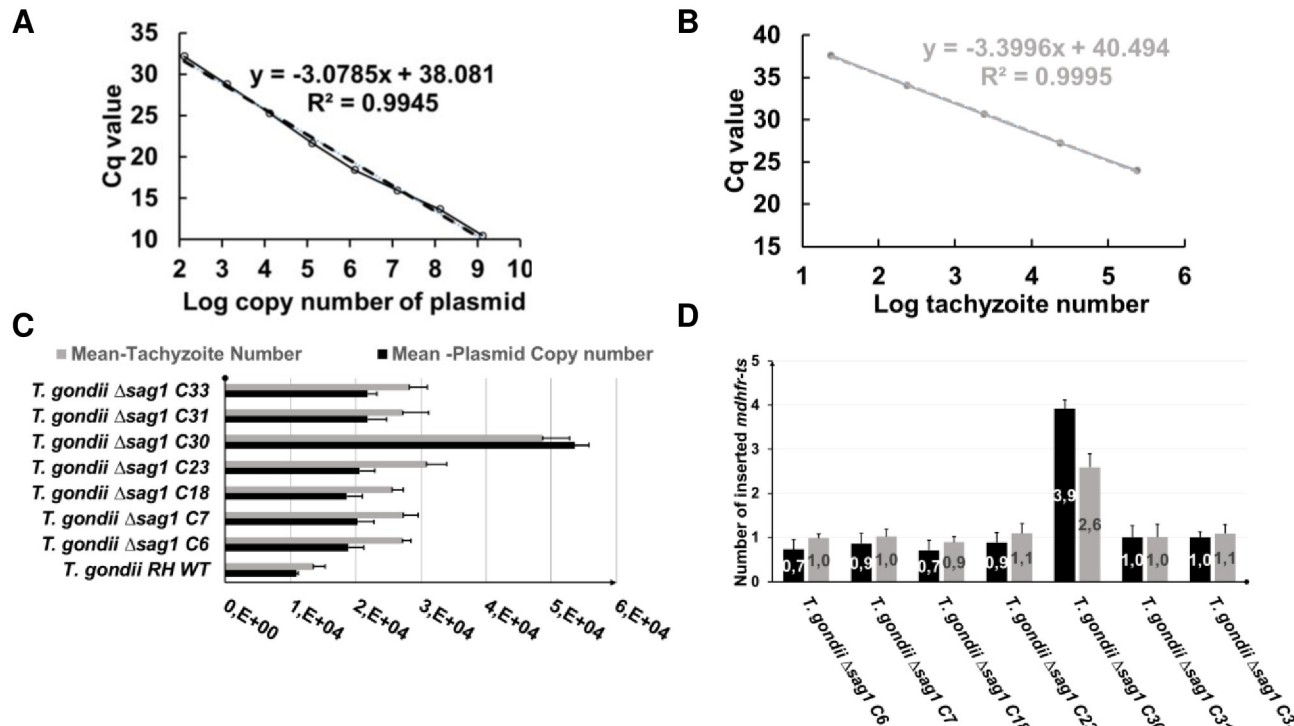

**Fig 4. Single TaqMan-qPCR for determining the copy number of integrated *mdhfr-ts* selectable marker.** Standard curves were made through a triplicate test of 10-fold serial dilutions of (**A**) P972 or (**B**) *T. gondii* RH DNA. (**C**) For each WT or KO clone, the number of existing *dhfr-ts* in the genome was determined according to the plasmid based standard curve (black bars) and the *T. gondii* RH DNA-based calibrator (grey bars). Since in the *T. gondii* genome the *wtdhfr-ts* is a single copy gene, the following equation was used: one WT tachyzoite = one-copy *dhfr-ts*, for the calculation based on *T. gondii* RH DNA based calibrator curve (grey bars). Error bars indicate standard deviation of triplicates for each sample. In (**D**), the number of inserted *mdhfr-ts* in each KO clone is defined by subtracting the *dhfr-ts* copy number found in the WT from the *dhfr-ts* copy number in the KO (black bars) or by subtracting the tachyzoite numbers determined for the WT from tachyzoite numbers corresponding the KO clone (grey bars). The optimal result of 1 indicates a single integration event of the *mdhfr-ts* into *sag1*.

TaqMan-qPCR enabled similar amplification efficiencies (R2 = 0.99%) for their respective targets (Fig 6A).

For the wild type DNA sample, the determined number of tachyzoites given by *dhfr-ts* amplification is equal to the number of tachyzoites determined by amplification of the *Toxoplasma* 529 bp sequence (Fig 6B), resulting thus in a ratio of 1 (Fig 6C). For a *T. gondii* RH Δ*sag1* clone having a single insertion of the *mdhfr-ts* within *sag1*, the calculated ratio is estimated to be 2, as it is the case for clone C6, 7, 18, 23, 31 and 33 (Fig 6C). For *T. gondii* RH Δ*sag1* C30, the number of tachyzoites given by *dhfr-ts* quantification was more than three times higher than the number of tachyzoites obtained by amplification of the *Toxoplasma* 529 bp repeat element (ratio > 3), which is indicative for multiple insertions of the *mdhfr-ts* fragment into the genome (Fig 6C).

## Southern blot analysis

To validate the results from both single and duplex TaqMan-qPCRs concerning the numbers of integrated *mdhfr-ts* fragments into the genome, Southern blot analysis of genomic DNA digested with BsaBI and DraIII was carried out (Fig 7). In the case of BsaBI digestion, (Fig 7A) the labeled probe recognized a 14.177-kb fragment in the *wt dhfr-ts* gene and a 4.999-kb fragment in the *mdhfr-ts* selectable marker integrated into *sag1*, such that the integrated fragment is easily identified in *sag1* KO parasites (Fig 7A). For genomic DNA digested with DraIII, the

- ## *T. gondii* RH (*wt*)

$$\frac{\text{Number of tachyzoites determined by amplification of } \textit{dhfr-ts}\ (\text{Cy5})}{\text{Number of tachyzoites determined by amplification of } \textit{T. gondii}\ \text{529 bp repeat element (FAM)}} = 1$$

- ## *T. gondii* RH Δ*sag1* (*single insertion of mdhfr-ts*)

$$\frac{\text{Number of tachyzoites determined by amplification of } \textit{dhfr-ts}\ (\text{Cy5})}{\text{Number of tachyzoites determined by amplification of } \textit{T. gondii}\ \text{529 bp repeat element (FAM)}} = 2$$

- ## *T. gondii* RH Δ*sag1* (*multiple insertion of mdhfr-ts >1*)

$$\frac{\text{Number of tachyzoites determined by amplification of } \textit{dhfr-ts}\ (\text{Cy5})}{\text{Number of tachyzoites determined by amplification of } \textit{T. gondii}\ \text{529 bp repeat element (FAM)}} > 2$$

**Fig 5. Principle and potential outcome of the duplex TaqMan-qPCR.**

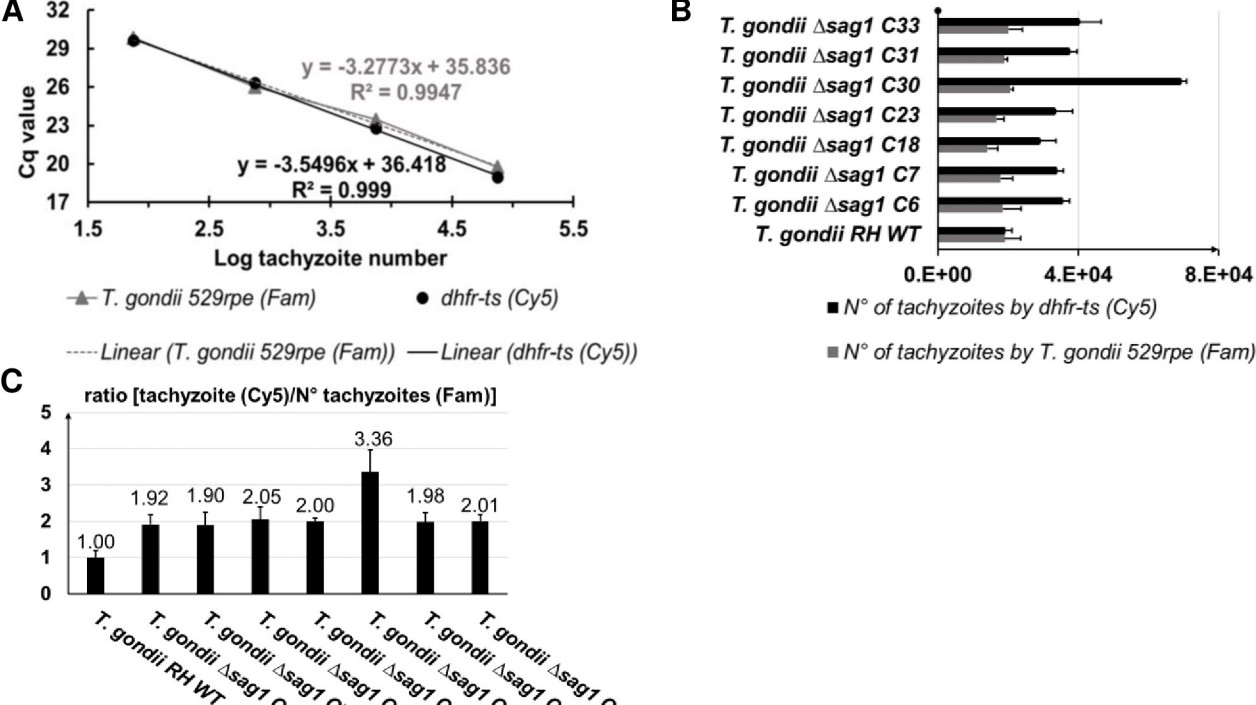

**Fig 6. Duplex TaqMan-qPCR for determining copy numbers of integrated *mdhfr-ts* selectable marker.** (**A**) Standard curve was made by using a 10-fold serial dilution of *T. gondii* RH DNA, with tachyzoites numbers ranging from 75 to $7.5 \times 10^5$ parasites. (**B**) For each WT or KO clone, the numbers of tachyzoites in the 3 ng DNA was determined according to amplification of *dhfr-ts* (black bars) and to the *T. gondii* 529 bp repeat element (grey bars). In (**C**), the number of inserted *mdhfr-ts* is given by the ratio of the number of tachyzoites as determined by *dhfr-ts* amplification and the number of tachyzoites determined by using the *T. gondii* 529 bp repeat element. A ratio equal to 2 indicates a single integration event of the *mdhfr-ts* in *sag1*. Error bars indicate standard deviations of triplicates for each sample.

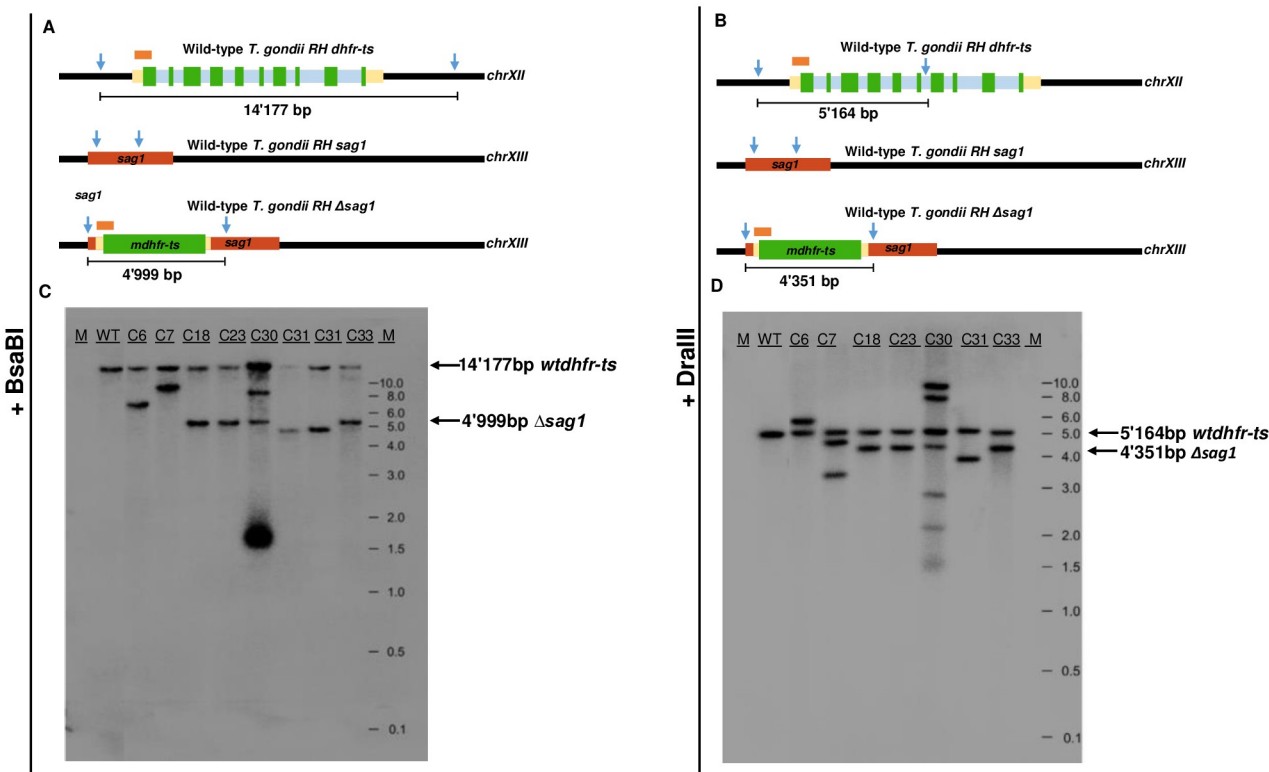

**Fig 7. Southern blot analysis for determining the number of mdhfr-ts integration events into the *T. gondii* RH genome.** (**A**) and (**B**) Schematic drawing of hybridization probe and restriction sites of BsaBI and DraIII in the WT *T. gondii* RH *dhfr-ts* gene and in the WT and mutant *sag1* locus. (**C**) Southern blot of genomic DNA digested with BsaBI and (**D**) with DraIII. M indicates the size of the fragments separated by gel electrophoresis.

*wt dhfr-ts* is present in all clones at 5.164 kb, and the integrated *mdhfr-ts* fragment within *sag1* is found at 4.351 kb (Fig 7B).

SB showed that WT *T. gondii* RH, as well as all seven KO clones, exhibited a single band corresponding to the *wt dhfr-ts* gene, migrating at 14.17 kb in BsaBI-digested DNA and at 4.99 kb in DraIII-digested DNA. Clones *T. gondii* RH Δ*sag1* C18, 23 31 and 33 exhibited a two-band pattern after digestion with BsaBI or DraIII, confirming thus a single integration event of the *mdhfr-ts* selection marker in the genome (Fig 7C and 7D). The band hybridizing with the probe in DNA of clone *T. gondii* RH Δ*sag1* C31 was at a lower position than the one observed for C18, 23 and 33. Thus, in agreement with the sequencing analysis, the inserted selectable marker within *sag1* in clone C31 is a truncated version of *mdhfr-ts*. For *T. gondii RH* Δ*sag1* C6, SB also revealed an integration of only one *mdhfr-ts* copy into the genome, but at another position than the *sag1* gene. This was also the case for *T. gondii RH* Δ*sag1* C7, with the exception that after genomic DNA digestion with DraIII, three bands were found to be hybridizing with the probe. Concerning the clone *T. gondii RH* Δ*sag1* C30, additional hybridizations were detected after digestion with BsaBI (two bands) or DraIII (four bands) besides the expected *wt dhfr-ts* and *mdhfr-ts* bands, indicating random and multiple integrations of *mdhfr-ts* into the *T. gondii* RH genome.

## Discussion

In this study, we have established a single- and duplex TaqMan-qPCR assay for determination of copy numbers of integrated *mdhfr-ts* selectable marker to evaluate of *T. gondii* RH KO

parasites generated by CRISPR-Cas9 as exemplified by using the major tachyzoite surface antigen TgSAG1 as KO target gene. *T. gondii* RH Δ*sag1* clones lacking the expression of TgSAG1 generated through CRISPR-Cas9-mediated KO were selected by treatment with Pyr, and the lack of TgSAG1 expression was ascertained by IFA and WB. Considering the risks of OTEs and thus the random integration of gene fragments into the genome, the *sag1* locus in different clones was amplified by PCR and respective fragments were sequenced to assess integration of the *mdhfr-ts* selection marker. A single- and duplex Taq Man qPCR for determination of the copy numbers of *mdhfr-ts* in *T. gondii* RH Δ*sag1* tachyzoites was developed, and was validated by SB.

Efficiency of gene editing in WT *Toxoplasma* using CRISPR-Cas9 (15%) was higher comparing to a frequency of $2 \times 10^{-5}$ obtained by non-homologous recombination [59]. The efficiency (15%) obtained herein can be considered satisfactory since WT *Toxoplasma* strains are significantly more relevant for studying gene function than most commonly used NHEJ-deficient Δ*ku80* strains. So far, frequency, severity, and the types of DNA sequence changes that might occur in association with the lack of NHEJ in Δ*ku80* parasites remains largely unknown. In apicomplexan parasites particularly *Theileria parva*, *Cryptosporidium spp*. and *Plasmodium spp*., loss of the classical NHEJ (C-NHEJ) pathway over genome evolution is suggested to be associated with reduced genome size (8–23 megabytes), this in comparison to the *T. gondii* genome (87 megabytes) that encodes the three main components of the C-NHEJ namely Ku70, Ku80 and DNA ligase IV [60]. In eukaryotic cells, impaired DNA-DSB repair pathways contributes to significant stress-induced effects and causes genomic instability [13, 14, 61]. Moreover, the use of Δ*ku80* strains for functional genomics does not prevent hazardous insertion of exogenous donor DNA. For example, cases of random integration into the genome were reported during reverse genetics in malaria parasites [62] naturally lacking key NHEJ compounds [63, 64].

For CRISPR-Cas9, OTEs resulting from non-targeted DNA mutations (base substitutions, deletions and insertions) are of low probability; in hematopoietic stem- and progenitor cells, the rates of insertion–deletion mutations did not differ between Cas9-treated and non-Cas9-treated cells [65]. These results were reported from two independent experiments targeting two different genes located in different chromosomes [65]. Thus, for reliable transgenesis and genome editing in *Toxoplasma* using selectable markers, selection protocols of engineered cells must include a step for determining whether an unintended integration of exogenous DNA has occurred.

Despite Southern blot analysis is ranked second after the WGS as the most unambiguous method for estimation of copy number in transgenic unicellular protozoan parasites, it has also significant disadvantages. Particularly, it is unsuitable for automation since the choice of restriction enzymes and probes are experiment-specific. Furthermore, digestion with restriction enzymes may result in DNA fragments larger than 15 kb, which are inefficiently blotted, leading thus to an underestimated copy number.

In contrast to SB, qPCR can be used to scan the entire genome for the presence of a selectable marker independently of the genomic location, and this can be done at higher throughput and in a wide dynamic range, which in turn allows simultaneous testing large numbers of samples in a short time frame. Consequently, qPCR was successfully implemented as an alternative to SB for characterization of transgene copy number and integration site in many different transgenic plant and animal cells [66, 67]. In this study, the strong evidence in line with this recommendation is *T. gondii* RH Δ*sag1* C30, which would have been taken for a correct mutant without further evaluation by single- and duplex TaqMan-qPCR, which detected multiple insertions. In addition, results for *T. gondii* RH Δ*sag1* C6 and C7 clearly demonstrate that both TaqMan-qPCRs can provide an absolute quantification of the inserted selection marker,

independently of its location in the genome. This was in line with PCR-Sequencing and SB findings, demonstrating a single copy integration of *mdhfr-ts* elsewhere in the genome for both *T. gondii* RH Δ*sag1* C6 and C7.

Concerning KO C7, the appearance of two bands in SB upon digestion with DraIII, but not with the BsaBI restriction enzyme, together with the results of the qPCRs, strongly suggest that the insertion of the single copy *mdhfr-ts* in an unknown genomic location has generated a new cutting site for DraIII.

Regarding the quantification of inserted *mdhfr-ts* copies in the examined clones, results obtained with the single TaqMan-qPCR were in correlation with those resulting from duplex TaqMan-qPCR. Thus, both single and duplex TaqMan-qPCR protocols can be applied as described herein each time *mdhfr-ts* is chosen as a selection marker in *Toxoplasma* gene KO experiments. So far, *mdhfr-ts* has been the most commonly used selection marker for transgenic *T. gondii* and *P. falciparum* [20, 68].

The duplex TaqMan-qPCR presented here can also be employed in case other selection markers are chosen. In such cases, primers and probes specific to the amplification of the *Toxoplasma* 529 bp repeat element can be used as reported here, however new primers and a TaqMan probe specific to the exogenous DNA needs to be designed. Subsequently, two important aspects need to be considered: (i) both primer sets must result in similar amplification efficiency and (ii) the standard curves must be made using *Toxoplasma* parasites as reference that have only one copy of the designed selection marker. Positive selection strategies based on drug resistance are limited in *T. gondii*, thus besides the *mdhfr* resistance gene [20] choices are almost restricted to *E. coli* chloramphenicol acetyl transferase (*cat*) [69, 70] or *Streptoalloteichus* ble (*ble*) [71] genes, which confer resistance to chloramphenicol or phleomycin, respectively. In order to ensure the maximum accuracy of single and duplex TaqMan-qPCR results, standardized protocols for cell-culture, tachyzoite purification, DNA extraction and quantification should be applied to all tested mutants / clones.

In conclusion, we have developed and validated sensitive, rapid and reliable single and duplex TaqMan qPCR methods for measuring *mdhfr-ts* copy numbers during CRISPR-Cas9 mediated gene editing in *Toxoplasma*. A significant advantage of these quantitative assays, particularly the duplex TaqMan qPCR, is that they can be easily applied for any selection cassette other than *mdhfr-ts*. Therefore, both qPCR techniques could become methods of choice for characterizing transgenic *T. gondii* cell-lines in term of integration pattern of the used exogenous DNA. Furthermore, by providing such a versatile molecular tool for quantitative detection of the integrated selection cassette, WT *T. gondii* stains can now be more frequently used instead of *ku80* KO strains.

## Supporting information

**S1 Raw images.**
(PDF)

**S1 Raw data.**
(XLSX)

## Acknowledgments

Anti-SAG1 and anti-IMC1 antibodies used in this study were a kind gift from Prof. Dominique Soldati-Favre, University of Geneva.

## Author Contributions

**Conceptualization:** Kai Pascal Alexander Hänggeli, Andrew Hemphill, Ghalia Boubaker.

**Data curation:** Kai Pascal Alexander Hänggeli, Ghalia Boubaker.

**Formal analysis:** Kai Pascal Alexander Hänggeli, Ghalia Boubaker.

**Funding acquisition:** Andrew Hemphill.

**Investigation:** Kai Pascal Alexander Hänggeli, Ghalia Boubaker.

**Methodology:** Norbert Müller, Bernd Schimanski, Philipp Olias, Joachim Müller, Ghalia Boubaker.

**Supervision:** Andrew Hemphill, Ghalia Boubaker.

**Validation:** Kai Pascal Alexander Hänggeli, Ghalia Boubaker.

**Visualization:** Ghalia Boubaker.

**Writing – original draft:** Kai Pascal Alexander Hänggeli, Ghalia Boubaker.

**Writing – review & editing:** Andrew Hemphill, Norbert Müller, Bernd Schimanski, Philipp Olias, Joachim Müller, Ghalia Boubaker.

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
