## [Decision Letter · Decision Letter 0]

26 Jul 2022

PONE-D-22-17694Single- and duplex TaqMan-quantitative PCR for determining the copy numbers of integrated selection markers during site-specific mutagenesis in Toxoplasma gondii by CRISPR-Cas9PLOS ONE

Dear Dr. Hemphill,

Thank you for submitting your manuscript to PLOS ONE. After careful consideration, we feel that it has merit but does not fully meet PLOS ONE’s publication criteria as it currently stands. Therefore, we invite you to submit a revised version of the manuscript that addresses the points raised during the review process.

Minor revision is needed.

We look forward to receiving your revised manuscript.

Kind regards,

Yong Qi

Academic Editor

PLOS ONE

Journal Requirements:

3. PLOS ONE now requires that submissions reporting blots or gels include original, uncropped blot/gel image data as a supplement or in a public repository. This is in addition to complying with our image preparation guidelines described at https://journals.plos.org/plosone/s/figures#loc-blot-and-gel-reporting-requirements. These requirements apply both to the main figures and to cropped blot/gel images included in Supporting Information. If the manuscript is positively reviewed, we will ask the authors to provide any missing raw image data for blot/gel results when they submit their first revision. As part of your review, please ensure that figures reporting blot or gel images comply with the journal’s image preparation guidelines and that the original data are provided following the journal’s request.  If you have any questions or concerns about blot/gel figures or data for this submission, please email us at plosone@plos.org before issuing a decision letter.

Reviewers' comments:

Reviewer's Responses to Questions

**Comments to the Author**

1. Is the manuscript technically sound, and do the data support the conclusions?

Reviewer #1: Yes

Reviewer #2: Yes

2. Has the statistical analysis been performed appropriately and rigorously? 

Reviewer #1: Yes

Reviewer #2: N/A

3. Have the authors made all data underlying the findings in their manuscript fully available?

Reviewer #1: Yes

Reviewer #2: No

4. Is the manuscript presented in an intelligible fashion and written in standard English?

Reviewer #1: Yes

Reviewer #2: Yes

5. Review Comments to the Author

Reviewer #1: The authors described a methodology to follow the mutations generated by CRIS Cas 9, for this they invalidated the Toxoplasma SAG1 gene and then used qPCR duplex to demonstrate the insertion of mutations. In general, the authors describe appropriately their methodology and results. The results are convincing, and they discuss the reach and limits of the method. I have only some minor suggestions to improve clarity for readers:

- They put in discussion something that should be described in results: the rate of success of mutated clones achieved in 5 of 33 clones (15%)

- One of the most critical and important results is the demonstration that SAG1 expression was invalidated as it was showed by western blot and IFAT experiments, this should NOT be supplementary material, please include as Figure 1

- The formula showed in Figure 4 please show as text in the section of results

Reviewer #2: The work by Haenggeli et al aims to describe single or duplex qPCR for the quantification of a mutated version of DHFR integrated as a drug marker during CRISPR/Cas9 genome editing in Toxoplasma gondii.

The manuscript is well written and scientifically sound. The authors developed a technique that estimates in an accurate manner if a commonly used drug marker for positive selection during genetic depletions in the Toxoplasma model is correctly inserted in the intended genomic locus, as well as if there were any off target insertions of multiple drug markers.

The insertion of multiple drug markers is a rare event nowadays, specially due to a highly efficient and targeted CRISPR/Cas9 double stranded break - as shown by the authors, which yielded a single clone with that caveat out of 33 analyzed. Also, although multiple off target insertions are not desired, it usually does not have a phenotypic impact on the parasites, unless multiple other genomic editions are planned - and, still, it is unlikely to alter the planned outcome.

Nonetheless, the protocols described are truly useful for researchers of the area that may be concerned with such ‘side effects’ of genomic editing.

Minor comments:

- The authors write several lines on how Ku80 strains have positively impacted the field, allowing and increment in transfection efficiency in Toxoplasma. However, they use wild type RH strain tachyzoites in their study. They even argument that, with their technique, WT parasites could be used more frequently in genomic editing protocols instead of base strains with genomic depletion of the ku80 gene. Why? Are WT parasites known to generate multiple off target incorporations of the drug markers? To my knowledge, the advantage of transfecting ku80 KO parasites is the increased efficiency of the incorporation of foreign DNA, not avoiding multiple insertions of drug markers. Please indicate/discuss with proper references on those affirmatives or remove them from the text.

- Minor revision of the text is desirable to eliminate certain formatting errors (e.g. references on line 58; lack of an additional espace on line 231; parenthesis on line 329; and so on…)

- Lines 282-288: Paragraph should be revised for clarity: Sanger sequencing is not demonstrated; insertion of “short DNA sequence” or integration of mDHFR-ts elsewhere also is not proven properly.

---

## [Author Response · Author response to Decision Letter 0]

11 Aug 2022

We have done so in the response letter.

---

## [Decision Letter · Decision Letter 1]

25 Aug 2022

PONE-D-22-17694R1Single- and duplex TaqMan-quantitative PCR for determining the copy numbers of integrated selection markers  during site-specific mutagenesis in Toxoplasma gondii by CRISPR-Cas9PLOS ONE

Dear Dr. Hemphill,

Thank you for submitting your manuscript to PLOS ONE. After careful consideration, we feel that it has merit but does not fully meet PLOS ONE’s publication criteria as it currently stands. Therefore, we invite you to submit a revised version of the manuscript that addresses the points raised during the review process.

Too many language errors exist in the manuscript.The manuscript can't be accepted until these errors are revised.e.g.line 20, "quantitative qPCR";line 78, ";";lines 146-148;line 159  "0.112 x 107", "2μM", "5μM";line 162;line 165;line 169 "tachyzoites extracted" ;line 171-173;... ...==============================

We look forward to receiving your revised manuscript.

Kind regards,

Yong Qi

Academic Editor

PLOS ONE

Journal Requirements:

Reviewers' comments:

Reviewer's Responses to Questions

**Comments to the Author**

1. If the authors have adequately addressed your comments raised in a previous round of review and you feel that this manuscript is now acceptable for publication, you may indicate that here to bypass the “Comments to the Author” section, enter your conflict of interest statement in the “Confidential to Editor” section, and submit your "Accept" recommendation.

Reviewer #1: All comments have been addressed

Reviewer #2: All comments have been addressed

2. Is the manuscript technically sound, and do the data support the conclusions?

Reviewer #1: Yes

Reviewer #2: Yes

3. Has the statistical analysis been performed appropriately and rigorously? 

Reviewer #1: Yes

Reviewer #2: N/A

4. Have the authors made all data underlying the findings in their manuscript fully available?

Reviewer #1: Yes

Reviewer #2: Yes

5. Is the manuscript presented in an intelligible fashion and written in standard English?

Reviewer #1: Yes

Reviewer #2: Yes

6. Review Comments to the Author

Reviewer #1: (No Response)

Reviewer #2: (No Response)

7. PLOS authors have the option to publish the peer review history of their article (what does this mean?). If published, this will include your full peer review and any attached files.

Reviewer #1: **Yes: **Jorge Gomez-Marin

Reviewer #2: No

---

## [Author Response · Author response to Decision Letter 1]

30 Aug 2022

Thank you for your comments. 

We have gone over the manuscript and eliminated all typos. Changes are marked in yellow in the corresponding manuscript file.

Many thanks and best regards

Andrew Hemphill

---

## [Editor Report · Decision Letter 2]

4 Sep 2022

Single- and duplex TaqMan-quantitative PCR for determining the copy numbers of integrated selection markers  during site-specific mutagenesis in Toxoplasma gondii by CRISPR-Cas9

PONE-D-22-17694R2

Dear Dr. Hemphill,

We’re pleased to inform you that your manuscript has been judged scientifically suitable for publication and will be formally accepted for publication once it meets all outstanding technical requirements.

Kind regards,

Yong Qi

Academic Editor

PLOS ONE
---

## [Editor Report · Acceptance letter]

8 Sep 2022

PONE-D-22-17694R2 

Single- and duplex TaqMan-quantitative PCR for determining the copy numbers of integrated selection markers during site-specific mutagenesis in *Toxoplasma gondii* by CRISPR-Cas9 

Dear Dr. Hemphill:

I'm pleased to inform you that your manuscript has been deemed suitable for publication in PLOS ONE. Congratulations! Your manuscript is now with our production department. 

Kind regards, 

on behalf of

Dr. Yong Qi 

Academic Editor

PLOS ONE